# Could Children’s Myopization Have Been Avoided during the Pandemic Confinement? The Conjunctival Ultraviolet Autofluorescence (CUVAF) Biomarker as an Answer

**DOI:** 10.3390/biomedicines12020347

**Published:** 2024-02-01

**Authors:** Miriam de la Puente, Cristina Irigoyen-Bañegil, Aura Ortega Claici, Jorge González-Zamora, Valentina Bilbao-Malavé, Patricia Fernandez-Robredo, María Hernández, Jesús Barrio, Alfredo García-Layana, Sergio Recalde

**Affiliations:** 1Retinal Pathologies and New Therapies Group, Experimental Ophthalmology Laboratory, Department of Ophthalmology, Clinica Universidad de Navarra, 31008 Pamplona, Spain; mdelapuentec@unav.es (M.d.l.P.); cirigoyen@unav.es (C.I.-B.); aortegaclai@alumni.unav.es (A.O.C.); jgzamora@unav.es (J.G.-Z.); valenbilbao@gmail.com (V.B.-M.); pfrobredo@unav.es (P.F.-R.); mahersan@unav.es (M.H.); aglayana@unav.es (A.G.-L.); 2Department of Ophthalmology, Clínica Universidad de Navarra, 31008 Pamplona, Spain; jbarrio@unav.es; 3Navarra Institute for Health Research, IdiSNA, 31008 Pamplona, Spain; 4Faculty of Medicine, Universidad de Navarra, 31008 Pamplona, Spain; 5Department of Ophthalmology, Bellvitge University Hospital, 08907 Barcelona, Spain

**Keywords:** myopia, CUVAF, confinement, outdoors, COVID-19

## Abstract

Background: The objective of this study was to evaluate the association of the presence of conjunctival ultraviolet autofluorescence (CUVAF) with the level and progression of myopia and the impact of reduced sunlight exposure during the COVID-19 pandemic confinement (PC). Methods: A retrospective observational study was carried out using three cohorts, children (9–17 years old), young adults (18–25 years old), and adults (>40 years old) with myopia (≤0.75D) and at least three annual eye examinations (before and after PC). All participants underwent an automatic objective refraction and CUVAF area analysis. All the participants filled out a questionnaire regarding lifestyle and myopia history. Results: The 298 recruited participants showed that during the PC, children’s and young adults' myopia progression rate increased on average by −0.50 and −0.30 D/year, respectively, compared with the pre-pandemic level (*p* < 0.0001 and *p* < 0.01). A significantly greater progression was observed in those with low baseline myopia compared to those with moderate or high myopia (*p* < 0.01). CUVAF shows its protective effect associated with outdoor activity (OA) with regard to the age of onset of myopia and mean diopters (*p* < 0.01). In fact, although there were no differences in the increase in diopters between children with and without CUVAF during the PC, those who had CUVAF started with lower gains (−0.3 D/year) compared to those who did not (−0.5 D/year; *p* < 0.05). The myopia treatments (atropine drops, Ortho-K, and MiSight^®^ contact lenses) showed a reduction effect in myopic progression rate post-PC in comparison with non-treated children (*p* < 0.0001, *p* < 0.0001 and *p* < 0.01, respectively). Conclusions: The strict restriction of OA during PC led to the rate of myopia progression doubling among children and young adults. This progression occurred mainly in children with previously low myopia, and CUVAF, as a biomarker of OA, reflects its potential to provide benefits in the form of recommended behavioral changes to protect against the development of myopia.

## 1. Introduction

One of the characteristics that distinguishes humans from the rest of the primates is the greater amount of exposed sclera, since, although it is barely visible in them, in our case it clearly stands out, signaling to any observer the direction of our gaze. In this regard, it has an important influence on non-verbal communication and, as a consequence, on our capacity for social interaction [1]. However, this great advantage comes with a price to pay, resulting in a greater non-pigmented area of the globe exposed to external agents such as ultraviolet (UV) radiation. As a result, a higher prevalence of conjunctival changes can be seen due to the effect of sun exposure, such as conjunctival ultraviolet autofluorescence (CUVAF), pinguecula, or pterygium [2]. Unlike pinguecula and pterygium, CUVAF is believed to represent preclinical lesions of UV-induced conjunctival damage that may lead to changes in intra- and extra-cellular metabolites that are ultimately responsible for conjunctival autofluorescence, probably due to the corneal focusing effect of peripheral light [3,4,5] (Figure 1). It has been objectified in different studies as a biomarker related to outdoor activities (OAs), which have shown an inverse correlation between time spent outdoors and myopia [6].

In response to the coronavirus disease 2019 (COVID-19) outbreak, in most countries, a period of home confinement was declared in order to prevent the spreading of the infection, which lasted until June 2020. During this confinement, it is estimated that more than 220 million school-aged children and adolescents were confined to their homes [7]. In Spain, the first pandemic wave began in mid-March 2020 and lasted 3 months, requiring house confinement and a strict lockdown. During the summer of that year, the confinement measures decreased, followed by a second wave that began in mid-September 2020 and lasted until December 2020 [8,9]. This resulted in collateral effects that include a significant increase in screen time and other near tasks at the expense of less time spent doing OA, reduced physical activity, and social isolation.

These lifestyle changes are potentially associated with an increase in the degree of myopia in the general population, and, in fact, it has been noticed that home confinement due to COVID-19 appeared to be associated with a substantial myopic shift in children [7]. Younger children’s refractive status may be more sensitive to environmental changes than that of older children, given that they are in an important period for the development of myopia [7]. As we have already mentioned, time spent outdoors was one of the aspects most affected during the COVID pandemic confinement (PC); therefore, the aim of this study was to examine the impact of the COVID-19 quarantine on the progression of myopia and its relationship with environmental factors as determined by the presence of CUVAF and questionnaires.

## 2. Materials and Methods

### 2.1. Study Design and Ethics Approval

A retrospective observational study was carried out using a cohort of children (6–17 years old), and two other cohorts of adults. The progression of myopia during the same period was compared to that of young adults (18–25 years old) and middle-aged adults (>40 years old) with some degree of myopia (<−0.75 diopters). The project was approved by the Institutional Review Board and the Ethics Committee of the Clínica Universidad de Navarra (study code 2021.070). All procedures carried out conformed to the guidelines of the Declaration of Helsinki. All the subjects that were included in the study were fully informed of the purpose and procedures of the study and informed consent was obtained for all participants.

### 2.2. Inclusion and Exclusion Criteria

We included myopic patients aged between 6–51 years old from optical shops (Multiópticas) and clinics from 18 cities of Spain with at least three annual visits (two before March 2020 and one after June 2020) and no more than 14 months between each visit. The general exclusion criteria were astigmatism or anisometropia equal to or greater than 2.0 diopters (D), amblyopia, congenital myopia or myopia associated with another pathology (premature retinopathy or systemic syndromes or diseases), previous conjunctival surgery, or any conjunctival pigmented lesion that might hinder the presence of CUVAF.

The participants were classified according to their non-cycloplegic autorefraction into “Mild Myopia: −0.75 D to −3 D” (M1), “Moderate Myopia: −3.25 D to −5.75 D” (M2) and “High Myopia: ≤−6 D”(HM). It should be pointed out that subjects with axial lengths ≥26 mm were automatically included in the “HM” group regardless of their spherical equivalent (SE).

### 2.3. Data Collection

All participants underwent an automatic objective refraction (with and without cycloplegia), color fundus photographs, and CUVAF imaging with a UV light system and measured by Fiji/ImageJ 1.6v (NIH, Bethesda, MD, USA). The CUVAF (-) value is given when there is no autofluorescence in any of the areas (nasal/temporal) of any of the participant's eyes, while the CUVAF (+) value is given when there is an autofluorescent spot measured by the Fiji tool in any of the analyzed areas of the participant.

Two of those refraction measurements were obtained before March 2020 and the third one after June 2020. Moreover, all subjects were asked to complete a questionnaire about lifestyle habits and myopia history.

In contrast to the participants recruited in the clinic, those who were recruited from the optic shops did not undergo cycloplegia. However, no differences were observed in the myopia progression rate of the two groups regardless of cycloplegic effect (Appendix A).

From the subtractions of the SE of visit 2 (V2)–visit 1 (V1) and visit 3 (V3)–V2, we obtained the rate of myopia progression in D/year of pre-PC and post-PC, respectively. From the subtraction of both, we obtained the difference in the rate of myopia progression due to PC (Figure 2).

### 2.4. Statistical Analyses

All the information was stored in accordance with data protection laws and grouped into variables in a database for subsequent analysis. The normality of all the variables was ensured by D’Agostino and Pearson testing.Variables were analyzed by Student’s *t*-test and one-way ANOVA, and Fisher’s F test was used for categorical variables. Corrected *p* values < 0.05 (two-tailed) were considered statistically significant using SPSS 20.1 (SPSS Inc., Chicago, IL, USA) and GraphPad Prism v 8.0.1 (GraphPad Software Inc., San Diego, CA, USA).

## 3. Results

### 3.1. Characteristics of the Study Sample

The total sample comprised 298 patients distributed across three age groups: 202 children (6–17 years), 80 young adults (18–25 years), and 16 middle-aged adults (over 40 years old). We collected their demographic, lifestyle, and myopia history data (Table 1). The questionnaire data showed that the percentage of participants with >30 h a week of near work activity (NA) (77%, 82%, and 50%, respectively; *p* < 0.01) and with >10 h a week of OA (32%, 38%, and 62.5%, respectively; *p* < 0.05) was different between the three study groups, with lower values in OA as age decreased. Likewise, during the PC, the three cohorts showed a decrease in OA by similar high percentages (78%, 73%, and 65%, respectively) and similar NA increases (80%, 83%, and 75%).

### 3.2. Association between Baseline Refractive Error and Outdoor Activities (Measured by CUVAF and Questionnaires)

Significative lower degrees of myopia were observed in patients with CUVAF in comparison to those without CUVAF in children and young adults (Figure 3A,B). The average refractive error at the beginning of the study was −4 D in CUVAF (-) children and −2.1 D in CUVAF (+), with a significant difference between groups of 1.9 D (*p* < 0.001) (Figure 3A). This difference persisted in the group of young adults, where CUVAF (-) patients had −4.2 D and CUVAF (+) patients had −3.1 D, resulting in a difference between groups of 1.1 D (*p* < 0.05) (Figure 3B). There were no significant differences in the group of middle-aged adults (*p* > 0.05).

When it comes to the baseline refractive error and lifestyle habits reported in the questionnaires for the children group, it was −3.7 D/year in the <10 h/OA group and −3.6 D/year in the >10 h/OA group (a non-significant difference of −0.1 D/year, *p* > 0.05) (Figure 3D). Similarly, in the group of young adults, it was −4.1 D/year in the <10 h/OA group and −4 D/year in the >10 h/OA group (a non-significant difference of −0.1 D/year, *p* > 0.05) (Figure 3E). There were no significant differences in the group of middle-aged adults (*p* > 0.05).

There was a statistically significant difference between the age of diagnosis of myopia between CUVAF (+) and CUVAF (-) patients (9 vs. 7 years, *p* < 0.05) (Figure 3C). On the other hand, there was no such difference between the <10 h/OA and >10 h/OA groups (8 vs 8 years, *p* > 0.05) (Figure 3F).

### 3.3. Rate of Myopia Progression

The evaluation time between the three visits remained constant; there were no significant differences (*p* > 0.05) in the time between pre-PC (V1 and V2; 12.8 months) and post-PC (V2 and V3; 12.9 months) visits. The difference in the rate of myopia progression during the COVID-19 PC affected all patients under 25 years old. The difference in the rate of progression among children was −0.5 D/year (*p* < 0.0001) during that period (pre-PC progression rate −0.5 D/year and post-PC progression rate −1.0 D/year; Figure 4A). For young adults, the change was −0.3 D/year (pre-PC progression −0.3 D/year and post-PC progression −0.6 D/year; *p* < 0.01; Figure 4B). Conversely, the middle-aged group experienced a non-significant change in D of +0.1 D/year (pre-PC progression −0.2 D/year and post-PC progression −0.1 D/year; *p* > 0.05; Figure 4C).

### 3.4. Difference in Myopia Progression Rate in Children and Young Adults after COVID-19 Confinement in Relation to Outdoor Activities (Measured by CUVAF)

The comparison of the difference in the rate of myopia progression after PC regarding the presence or absence of CUVAF showed no significant differences in the children group, with a difference in the rate of myopia progression of −0.5 D/year for CUVAF (+) and −0.5 D/year for CUVAF (-) (*p* > 0.05; Figure 5A). Similarly, in the group of young adults, there was no significant difference observed, with −0.4 D/year for CUVAF (+) and −0.6 D/year for CUVAF (-) (*p* > 0.05) (Figure 5B).

However, when pre- and post-PC progression was analyzed, CUVAF (-) children presented significantly higher rates of progression compared to CUVAF (+) children (*p* < 0.01 and *p* < 0.05, respectively, Figure 5C). In the group of young adults, however, no significant differences were observed (*p* > 0.05; Figure 5D).

### 3.5. Myopia Progression Rate after COVID-19 Confinement Regarding Myopia Classification and Outdoor Activities (Measured by CUVAF) in Children

The greatest difference in myopia progression rate after PC was seen in the M1 group (−0.7 D/year gain), followed by the M2 group (−0.4 D/year gain) and the HM group (−0.3 D/year gain) (*p* < 0.01, Figure 6A). When the pre-PC myopia progression rate was analyzed, a great difference between the three groups was observed (*p* < 0.0001), with M1 showing the lowest rate of pre-PC progression and HM the highest. However, in the post-PC progression rate analysis, these differences disappeared between M1 and M2 (*p* > 0.05) and were less notable between M1 and HM (*p* < 0.05; Figure 6B).

In each group, the percentages of children with CUVAF (+) and (-) were analyzed, and a higher percentage of children from the M1 group were shown to be CUVAF (+) compared with the M2 and HM groups (*p* < 0.001; Figure 6C). The analysis of the difference in the myopia progression rate after PC among the M1, M2, and HM groups demonstrated that M1 CUVAF (-) children had higher differences in progression rates in comparison to M1 CUVAF (+) children; in all degrees, the CUVAF (+) (*p* < 0.05). No differences were found among M2 and HM groups (Figure 6D).

### 3.6. Difference in Myopia Progression Rate after COVID-19 Confinement Regarding Different Types of Myopia Treatments in Children

Some of the children in the study were receiving different treatments to reduce myopia progression before and during PC. The difference in myopia rate progression between treated and non-treated children was analyzed. It was observed that atropine drops and Ortho-K contact lenses resulted in a mean difference in myopia progression rate of +0.6 D/year in both groups. When compared to non-treated children, children treated with atropine drops and Ortho-K contact lenses showed a difference in myopia progression rate of +1.1 D/year (*p* < 0.0001 and *p* < 0.001, respectively) (Figure 5). Children treated with MiSight^®^ contact lenses showed a +0.7 D/year difference in myopia progression rate compared to non-treated children (*p* < 0.01) (Figure 7).

## 4. Discussion

Myopia is the most common refractive error worldwide and is expected to increase in prevalence in the coming years [10]. Our understanding of how risk factors contribute to the increase in myopia is limited, and improving our knowledge could help us to implement preventive and therapeutic measures.

The objective of this study was to analyze the impact of confinement during the COVID-19 pandemic on myopia progression, and its correlation with environmental factors, measured by the presence of CUVAF and questionnaires. CUVAF has been linked to outdoor activity and sunlight exposure, suggesting it may be a conjunctival change due to cumulative UV radiation [11,12,13,14]. In this regard, previous studies have correlated the CUVAF area with lower myopia levels, suggesting a protective effect of sunlight exposure against the development of myopia [6,11,12,15]; our results are in line with this (Appendix A). A great advantage of CUVAF as a biomarker is its subacute nature, which in this case can help determine pre-PC outdoor behavior and analyze the effect of PC between groups with differences in their time spent doing OA.

Reduced OA is a well-known risk factor for myopia development and progression [11,16,17,18,19]. In Spain, the COVID-19 PC was very strict, with no OA allowed for several months, and then short daylight periods outside were permitted until the end of the confinement, with a subsequent increase in NA [8]. As seen in our study, this period of PC resulted in an increase in the rate of myopia progression among children and young adults; in fact, the rate of progression doubled, which gives us an indication of the impact of OA on myopia progression. This increase in the rate of myopia progression was also seen in CUVAF (-) children, who already spent less time outdoors before the pandemic, meaning that even minor changes in sunlight exposure can have a strong effect on myopia. On the other hand, no changes were seen in the middle-aged adults group; this aligns with findings in the literature indicating that younger individuals are more susceptible to shifts in myopia and more vulnerable to environmental changes [18,20].

The increase seen in the rate of myopia progression post-PC in children varied among different groups of myopia, with significantly greater progression observed in those with low baseline myopia compared to those with moderate or high myopia; in fact, the change in the rate of myopia progression was almost five times higher in that group. As a result, the rate of myopia progression post-PC reached similar values among the three myopia groups. This shows the impact that a sudden stop in OA can have on children with lower myopia, and it points out how important it can be to emphasize the promotion of OA in this group of patients. Moreover, these results can be explained by the fact that children with low myopia are the ones that spent more time doing OA pre-PC, as reflected by the CUVAF presence in this group in our results; however, when CUVAF (+) and (-) children were compared, only those with low myopia showed a significant increase in progression rate in the CUVAF (-) group. In addition, when the same analysis was performed by age groups, no differences were observed. All of these results highlight the role of OA in preventing the trigger of myopia progression in children with low myopia, while its effect is less crucial in groups with moderate and high myopia.

When comparing the objectiveness and reliability of CUVAF as a biomarker of OA to questionnaires about time spent outdoors, no differences were found in the data obtained from questionnaires in SE by group age, and age of myopia onset, in contrast with the significant differences obtained when performing the same analysis using CUVAF to measure OA. This demonstrates that CUVAF is an objective and measurable indicator of OA and has been proved to be more useful than subjective questionnaires, probably because of inaccurate reporting or recall bias [13].

Regarding myopia prevention, all the treatments analyzed in our study (atropine drops, Ortho-K, and MiSight^®^ contact lenses) showed a reduction in myopic progression rate post-PC in comparison with non-treated children, with atropine drops showing the greatest statistical significance. These results are of great importance because they demonstrated that even in children with zero OA and increased NA, which increase the myopic progression rate (−0.5 D/year), these preventive treatments overcome this deleterious effect and are even able to reduce the progression rate (+0.6 D/year for atropine drops). The protective effect of myopic-preventive therapies has been widely described in different studies [21,22], although this is the first time that the efficacy of these therapies has been evaluated in children under strict deprivation of OA and increased NA, which are the main environmental modifiable factors.

One of the main advantages of our study is the strict restriction of sunlight exposure, which was only possible due to the COVID-19 lockdown, and that allowed us to reach the previous results. However, these extreme conditions might be a limitation in efforts to extrapolate the effect of behavioral changes promoted in clinical practice in the progression of myopia, although the highly significant results suggest that they may be relevant to less strict situations and, therefore, could have higher external validity. Furthermore, although not all refraction measurements were conducted under cycloplegia, their validation with a sample of children under cycloplegia enhances the internal validity of our results (Appendix A). Another limitation is the retrospective nature of the study, which led to the absence of CUVAF data from prior to the PC. A prospective randomized trial would have made it possible to evaluate changes during this period in order to study the usefulness of CUVAF in monitoring short-term changes in OA, and to obtain a higher level of evidence. Unfortunately, events such as this pandemic cannot be predicted and the study had to be retrospective.

To our knowledge, this is the first study to demonstrate the important role of OA in reducing the myopia progression rate in children during the initial phases of myopia, with less relevance in more advanced stages; and the beneficial effect of myopia preventive treatments even in conditions of OA deprivation and increases in NA. Moreover, our study highlights the importance of age in myopia progression after the cessation of OA, with children and young adults being the more susceptible age groups, and also demonstrates the utility of CUVAF as a biomarker of OA and its usefulness in determining the population most likely to benefit from the promotion of a change in visual habits and increased time spent outdoors.

## 5. Conclusions

The strict restriction of OA during the pandemic led to the rate of myopia progression doubling among children and young adults. This progression occurred mainly in children with previously low myopia, which showed an increase of almost five times, with a less relevant effect in more advanced stages. Moreover, the use of myopia treatments was effective in reversing this acceleration of progression. Finally, the role of CUVAF as a biomarker of OA proves useful, and reflects its potential as a parameter for assessing the population with the greatest potential to benefit from behavioral changes to protect against the development of myopia.

## Figures and Tables

**Figure 1 biomedicines-12-00347-f001:**
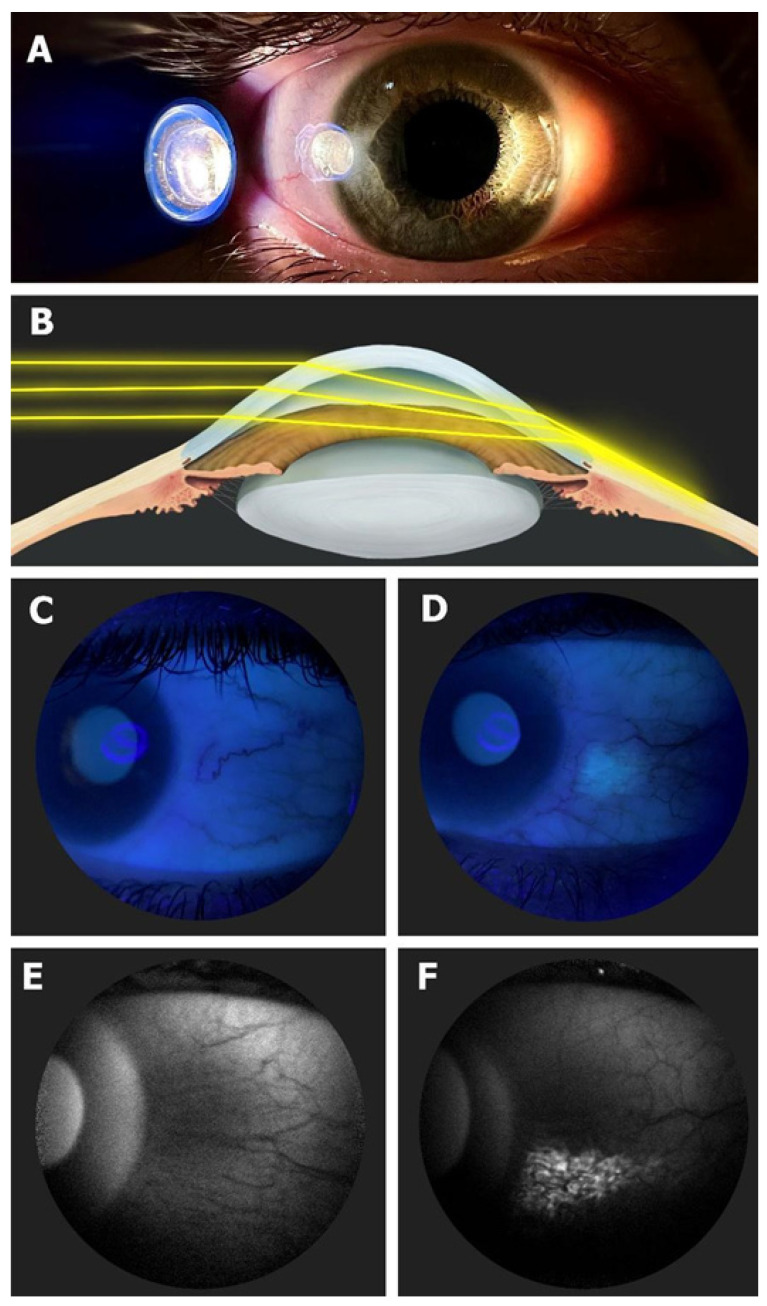
(**A**) In vivo demonstration of corneal focusing of peripheral light coming from the temporal side of the eye onto the limbus and nasal conjunctiva with higher light intensity compared to the temporal side. (**B**) Optical representation of the peripheral light-focusing effect that leads to a concentration of the incoming rays of light, passing through the anterior chamber, onto the contralateral limbo-conjunctival surface of the eye. (**C**) CUVAF negative (no conjunctival hyperautofluorescent area is seen) in a color photograph taken under UV light (peak wavelength of 365 nm). (**D**) CUVAF positive (demonstrates an area of hyperautofluorescence in the right nasal interpalpebral region) in a color photograph taken under UV light (peak wavelength of 365 nm). (**E**) CUVAF negative (no conjunctival hyperautofluorescent area is seen) in a photograph taken using the BAF mode of the Heidelberg Spectralis HRA + OCT (peak wavelength of 488 nm). (**F**) CUVAF positive (demonstrates a triangular conjunctival hyperautofluorescent area with limbal base and temporal apex) in a photograph taken using the BAF mode of the Heidelberg Spectralis HRA + OCT (peak wavelength of 488 nm). Image courtesy of Gutierrez-Rodriguez et al [6].

**Figure 2 biomedicines-12-00347-f002:**
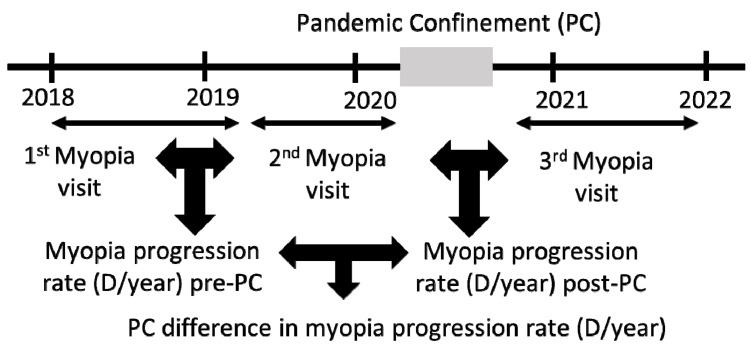
Calculation of myopia progression rate pre- and post-PC to obtain the difference in progression rate due to PC.

**Figure 3 biomedicines-12-00347-f003:**
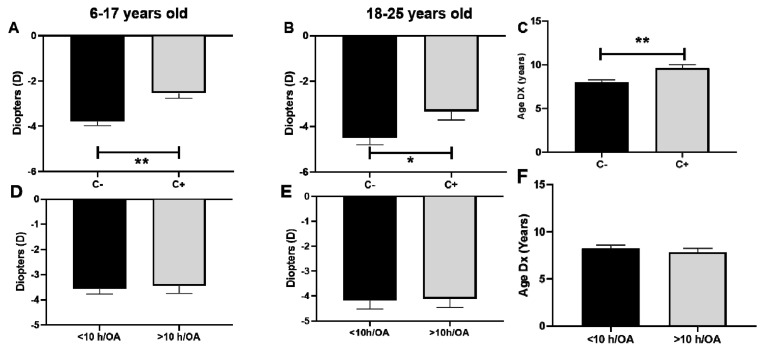
Association between baseline refractive error and outdoor activities measured by CUVAF (**A**–**C**) and questionnaires (**D**–**F**). (**A**,**D**): Children aged 6–17 years. (**B**,**E**): Young adults aged 18–25 years. (**C**,**F**): Association between age of myopia diagnosis and outdoor activity measured by CUVAF and questionnaires. C: CUVAF (conjunctival ultraviolet autofluorescence). h/OA: hours spent doing outdoor activity (weekly). DX: age of myopia diagnosis. Significance *p* < 0.05. * *p* < 0.05, ** *p* < 0.01.

**Figure 4 biomedicines-12-00347-f004:**
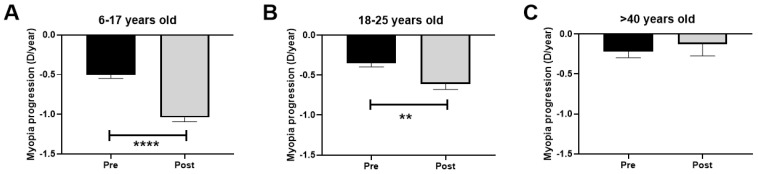
Rate of myopia progression (diopters/year) pre- and post-PC. (**A**): Children aged 6–17 years. (**B**): Young adults aged 18–25 years. (**C**): Middle-aged adults >40 years. PC: pandemic confinement. Significance *p* < 0.05. ** *p* < 0.01, **** *p* < 0.0001.

**Figure 5 biomedicines-12-00347-f005:**
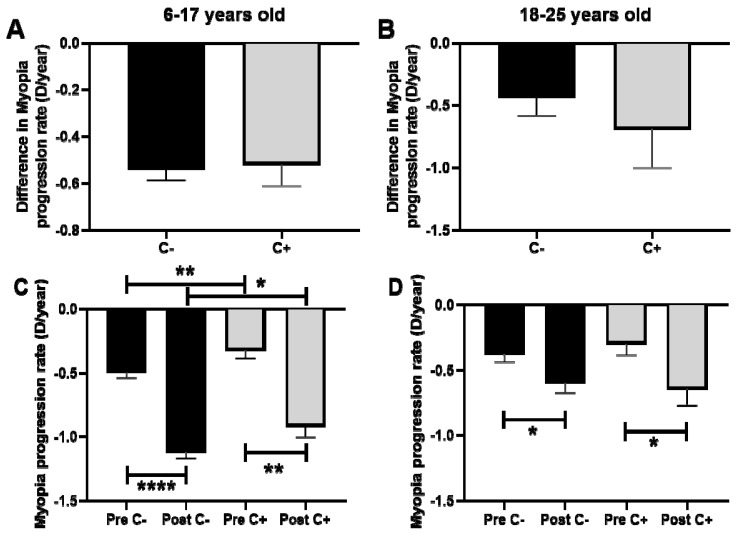
(**A**,**B**): Comparison of the difference in the rate of myopia progression after PC regarding the presence or absence of CUVAF. (**A**): Children aged 6–17 years. (**B**): Young adults aged 18–25 years. (**C**,**D**): Comparison of the myopia progression rate (D/year) in pre- and post-PC in relation to the presence of CUVAF. (**C**): children aged 6–17 years. (**D**): young adults aged 18–25 years. C: CUVAF (conjunctival ultraviolet autofluorescence). PC: pandemic confinement. Significance *p* < 0.05. * *p* < 0.05, ** *p* < 0.01, **** *p* < 0.0001.

**Figure 6 biomedicines-12-00347-f006:**
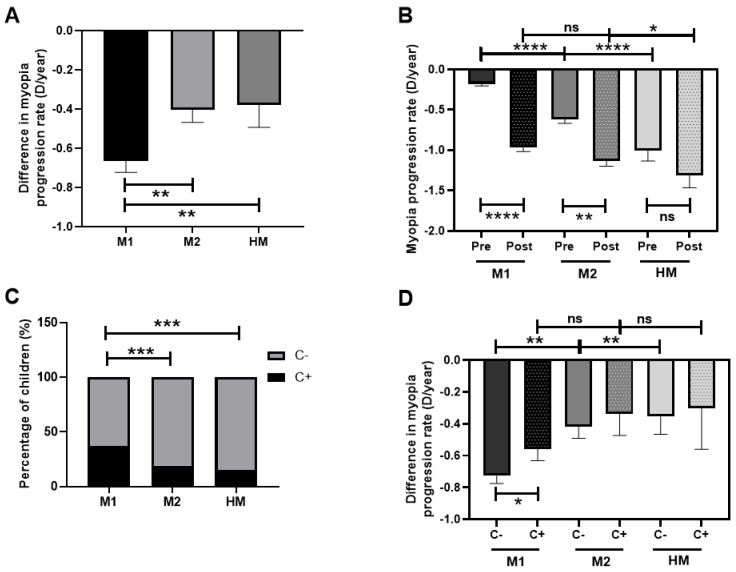
Myopia progression rate (D/year) after COVID-19 confinement regarding myopia classification and outdoor activity (measured by CUVAF) in children. (**A**): Difference in myopia progression rate after PC in M1, M2, and HM groups. (**B**): Myopia progression rate pre- and post-PC among M1, M2, and HM groups. (**C**): Percentages of CUVAF (+) and CUVAF (-) children in M1, M2, and HM. (**D**): Difference in myopia progression rate after PC among M1, M2, and HM groups and the presence of CUVAF. C: CUVAF (conjunctival ultraviolet autofluorescence). PC: pandemic confinement. Significance *p* < 0.05. * *p* < 0.05, ** *p* < 0.01, *** *p* < 0.001, **** *p* < 0.0001, ns; non-significant.

**Figure 7 biomedicines-12-00347-f007:**
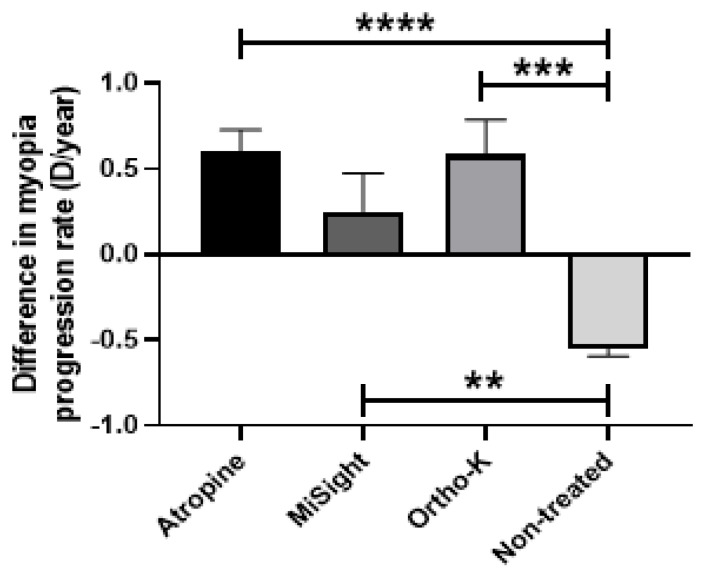
Differences in myopia progression rates after COVID-19 confinement regarding different types of myopia treatments in children. Atropine drops; MiSight^®^, and Ortho-K contact lenses. Significance *p* < 0.05. ** *p* < 0.01, *** *p* < 0.001, **** *p* < 0.0001.

**Table 1 biomedicines-12-00347-t001:** Demographic characteristics of participants. F: female. NA: near work activity. OA: outdoor activity. PC: pandemic confinement. ns: not significant. Significance *p* < 0.05. * *p* < 0.05, ** *p* < 0.01, *** *p* < 0.001.

	Total	6–17	18–25	>40	*p*
N (%)	298 (100)	202 (68)	80 (27)	16 (5)	
Gender (F) (%)	69%	66%	75%	75%	ns
Age (years)	19.8	13.9	21.5	44.4	***
Age of onset of myopia (years)	9.3	8.4	11.1	9.5	*
Diopters (D)	−3.96	−3.5	−4.05	−5.8	**
N° of high myopes (HM) (%)	18.2%	16%	27%	62.5%	***
>30 h of NA/week (%)	78.2%	77%	82%	50%	**
Increased NA during PC	81%	80%	83%	75%	ns
>10 h of OA/week (%)	38.2%	32%	38%	62.5%	*
Decreased OA during PC (%)	74%	78%	73%	65%	ns

## Data Availability

The raw data supporting the conclusions of this article will be made available by the authors upon request.

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
