# Peer review of "Could Children’s Myopization Have Been Avoided during the Pandemic Confinement? The Conjunctival Ultraviolet Autofluorescence (CUVAF) Biomarker as an Answer"

_biomedicines, 2024, doi:10.3390/biomedicines12020347_

Round 1

Reviewer 1 Report

Comments and Suggestions for Authors

This is a retrospective observational study looking at myopia progression before and after pandemic confinement in children and adults. The authors found that myopia progressed more with children and young adults, while older adults remained stable. They also assessed outdoor light exposure as measured through conjunctival autofluorescence in relation to myopia progression. There were no differences in myopia progression between children with and without increased outdoor light exposure, although those who had significant outdoor light exposure started with lower myopic progression.

These are all important finding and worthy of publication.

I have some specific comments:

LIne 15. We are here to "evaluate the association" not make an association. 

Line 19. I would not use the word "graduation," perhaps annual eye examinations would be better.

Line 39. The wording is awkward because, as we know, homo sapiens are primates. I would rephrase this.

Line 204. I would like to see a bit more analysis of Figure 5 in the discussion. What do you think is happening here? Basically, it looks like both children who didn't spend as much time outdoors and those that did experienced greater levels of myopia progression after pandemic confinement. I suppose that speaks to the strength of the effect of pandemic confinement.

Line 313. One weakness of the study is that probably most were non-cycloplegic and some cycloplegic exams, although you have attempted to address this concern.

Also, the study is susceptible to bias because it is retrospective and not a prospective randomized trial. This should be mentioned toward the end of the discussion. I still believe, however, that the results are worthy of publication.

In summary, this is a well designed study and well written manuscript. I recommend publishing with some minor edits.

Comments on the Quality of English Language

For the most part, aside from what I mentioned above, this is well written.

Reviewer 2 Report

Comments and Suggestions for Authors

This study was to investigate the presence of Conjuntival Ultraviolet Autofluorescence (CUVAF) with the progression of myopia and the impact of reduced sunlight exposure during the COVID-19 pandemic confinement. However thre are some questions should be clarified.

1.       In the abstract: CUVAF shows its protective effect associated with OA with regard to the age ….   OA?

2.          In the study`, CUVAF negative (no conjunctival hyperautofluorescent area is seen) and CUVAF positive (demonstrates a triangular conjunctival hyperautofluorescent area with limbar base and temporal apex). The major concern is that are there any care with borderline CUVAF? i.e. spot or weak hyperautofluorescent but not triangular conjunctival hyperautofluorescent area? In addition, how about the relationship between the size of hyperautofluorescent area and myopia progression?  

3.          Dose the reading time (book, cell phone, TV, PC….) affect the progression of myopia among these 3 groups>  

Reviewer 3 Report

Comments and Suggestions for Authors

This is an interesting manuscript investigating the role of the COVID-19 pandemic confinement in children myopia. During this confinement children were restricted in home and spent more time before screen either for school activities or entertainment at the expense of outdoor activities. This manuscript investigates the role of the Conjunctival Ultraviolet Autofluorescence (CUVAF) Biomarker as an answer for children myopization during the pandemic.

In fact the study showed that the confinement resulted in increase in the rate of myopia progression among both children and young adults (actually the rate of progression doubled), giving an indication of the impact outdoor activities on myopia progression.

The manuscript is well written and designed and appropriate for the journal.

 A few comments for improvement are:

1.       Table 1 is rather too short, should be removed and the content described in the relevant paragraph.

2.       The supplement figure (S1) as provided in the separate file require explanation of the terms “ns”ànon significant, and the statistical significance of the star symbols. A figure caption would be helpful. Propose to move the relevant information from the manuscript to the supplement file.

3.       The major concern for this manuscript relies in the statistical analysis paragraph, whereas is stated “Continuous variables were analyzed by Student’s t-test and one-way ANOVA” notably both tests are parametric, meaning that data normality within each category group should be ensured. Thus, please update this paragraph pointing to the normality tests applied (usually Kolmogorov Smirnov and Shapiro Wilk). In case that normality is not ensured by one of the previous tests, then the appropriate tests for analysis are non-parametric (such as Mann Whitney U – test instead of t-test and Kruskal Wallis instead of ANOVA). This reviewer believes that the study results will not change even if non parametric tests will be applied, actually is expected that smaller p-values will be obtained.

Minor

1.       Line 141 seems that there is an extra parenthesis

2.       Is there a minus sign missing from Diopters (D) à3.96 in table 2?

Round 2

Reviewer 2 Report

Comments and Suggestions for Authors

the authors have answer my comments